# Personality and psychopathological characteristics in functional movement disorders

**Antonina Luca**[1], **Tiziana Lo Castro**[2], **Giovanni Mostile**[2,3], **Giulia Donzuso**[2], **Calogero Edoardo Cicero**[2], **Alessandra Nicoletti**[2], **Mario Zappia**[2]*

**1** Department of Medicine and Surgery, Kore University of Enna, Enna, Italy, **2** Department of Medical, Surgical Sciences and Advanced Technologies, GF Ingrassia, University of Catania, Catania, Italy, **3** Oasi Research Institute-IRCCS Troina, Troina, Italy

* m.zappia@unict.it

## Abstract

### Introduction

Aim of the present study was to assess personality and psychopathological characteristics in patients with functional movement disorders (FMDs) compared to patients with other neurological disorders (OND).

### Methods

In this cross-sectional study, patients affected by clinically established FMDs and OND who attended the Neurologic Unit of the University-Hospital "Policlinico-San Marco" of Catania from the 1st of December 2021 to the 1st of June 2023 were enrolled. Personality characteristics were assessed with the Rorschach test coded according to Exner's comprehensive system and the Structured Clinical Interview for DSM-5 (SCID-II).

### Results

Thirty-one patients with FMDs (27 women; age 40.2±15.5 years; education 11.7±3.2 years; disease duration 2.3±2.5 years) and 24 patients affected by OND (18 women; age 35.8 ±16.3 years; education 11.9±2.9 years; disease duration 3.4±2.8 years) were enrolled. At the Rorschach, FMDs presented a significantly higher frequency of Popular (P) and sum of all Human content codes (SumH>5) responses and avoidant coping than OND.

### Conclusion

FMDs presented "conformity behaviors", excessive interest in others than usual a maladaptive avoidant style of coping and a difficulty in verbalizing emotional distress. These psychopathological characteristics may favor the occurrence of FMDs.

**Data Availability Statement:** All relevant data are within the manuscript and its Supporting Information files.

**Funding:** The author(s) received no specific funding for this work.

**Competing interests:** No authors have competing interests

## Introduction

Functional movement disorders (FMDs) are movement disorders that cannot be explained by typical neurological diseases or other medical conditions [1]. FMDs are a frequent cause of disability and poor quality of life and have a notable economic impact due to loss of employment, inappropriate or unnecessary diagnostic procedures and delayed diagnosis [2]. From a phenotypical point of view, FMDs can present with any type of movement disorder, including weakness, tremor, dystonia, gait disorders, myoclonus and chorea, often in combinations [3]. The treatment of FNDs frequently requires a non-pharmacological approach based on both psychotherapy and physiotherapy, meticulously tailored to the patients and requiring a multidisciplinary clinical approach [4,5].

Although the exact mechanisms leading to FMDs are still unclear, it has been emphasized that predisposing vulnerabilities (i.e. personality traits, psychiatric comorbidities, alexithymia. . .), precipitants (i.e traumatic events, post-traumatic stress disorder. . .) and perpetuating factors (i.e family and/or financial concerns, lack of social support. . .) exert a pivotal role in FMDs occurrence [6]. However, to date, the role of personality traits possibly associated with the occurrence of FMDs still need to be clarified [7,8].

Yet, differently from mood states, personality is an essentially stable internal factor making individual behavior consistent over time. Moreover, personality traits exert a fundamental role on stress vulnerability, coping effectiveness and human interactions [9]. Hence, the individualization of peculiar personality traits may be useful for a tailored approach during the diagnosis communication and the choice of therapeutic options.

Aim of the present study was to assess personality and psychopathological characteristics in patients with FMDs compared to patients with other neurological disorders (OND) as control group.

## Materials and methods

This cross-sectional study was performed according to The Reporting of studies Conducted using Observational Routinely-collected health Data (RECORD) Statement.

Patients affected by laboratory supported FMDs [10] who attended the Neurologic Unit of the University-Hospital "Policlinico-San Marco" of Catania from the 1st of December 2021 to the 1st of June 2023 were enrolled in the study and compared to patients affected by OND. Patients with cognitive impairment were excluded from the study. This study was performed in line with the principles of the Declaration of Helsinki and has been approved by the Ethical Committee of the University Hospital "Policlinico-San Marco" of Catania, Italy. Protocol number: 117/2021/PO. All the enrolled subjects signed the consent to participate.

### Instruments

**Neuropsychological assessment.** All the enrolled subjects underwent the following comprehensive neuropsychological battery in order to exclude patients with cognitive deficits: global cognition screening test (Mini Mental State Examination, Montreal Cognitive Assessment), episodic memory (Rey's Auditory Verbal Learning Test), attention (Stroop color-word test), executive functioning (Verbal fluency letter test, Frontal Assessment Battery) and visuospatial functioning (Clock drawing test).

The presence of depression and anxiety were evaluated with, respectively, the Hamilton Depression Rating Scale (HDRS) and the State Trait Anxiety Inventory (STAI Y1-Y2).

**Personality assessment.** Personality characteristics were assessed with the *Rorschach test* coded according to Exner's comprehensive system [11] and administrated by a psychologist with certified expertise (TLC). The Rorschach consists of 10 inkblots, administered and coded

in a standardized way. Seven major clusters and their related variables were evaluated: 1) control and stress tolerance, 2) affect, 3) ideation, 4) cognitive mediation, 5) information processing, 6) self-perception, 7) interpersonal perception (**S1 Table**).

The presence of Personality Disorders (PeDs) diagnosed according to the Diagnostic and Statistical Manual for Mental Disorders 5 edition (DSM-5) was assessed through the *Structured Clinical Interview for DSM-5 (SCID-II)*.

### Statistical analysis

Due to the lack of studies assessing personality with the Rorschach test in patients with FMD, the study sample size was calculated considering the prevalence of PeDs assessed with the SCID-II [16]. Assuming an alpha error level of 0.05 and a power of 80%, to reach a significant difference (p<0.005), the minimum number of participants required was 16 in FMD group and 16 in the OND group, for a total of 32 patients.

Data were analyzed using STATA 16 software packages (StataCorp, College Station, TX, United States). Quantitative variables were described using mean and standard deviation. The Shapiro–Wilk normality test was performed. Differences between means were evaluated with the unpaired t-test in the case of normal distribution and the Mann–Whitney U test for not-normal distribution. Qualitative variables were described using frequency and percentage and compared with the Chi square test. Univariate analysis was performed to assess the associations between demographic and psychological characteristics and FMD, considered as outcome variable. Age and sex considered a priori confounders and variables with p-value< 0.1 at univariate analysis were included in the multivariate analysis. The significance level was set at 0.05 and 95% Confidence Intervals (CI) were calculated. A statistical analysis considering FMD phenotypes was finally performed.

### Results

Thirty-one patients with FMDs (27 women; age 40.2±15.5 years; education 11.7±3.2 years; disease duration 2.3±2.5 years) and 24 patients affected by OND (18 women; age 35.8±16.3 years; education 11.9±2.9 years; disease duration 3.4±2.8 years) were enrolled. Among FMDs group, the predominant symptom was weakness (w), variously associated with tremor (FMD-wT, n.8; 25.8%) and gait disorders (FMD-wGD, n.23; 74.2%). Considering FMD phenotype, no statistically significant differences were found comparing FMD-wT and FMD-wGD in terms of sex [respectively 7 (87.5%) *versus* 20 (86.9%) women, p-value 0.968], age (respectively 37.5±14.8 *versus* 41.2±15.9 years, p-value 0.568) and education (respectively 10.8±3.6 versus 12.0±3.13 years, p-value 0.406).

The OND group was represented by patients with multiple sclerosis (n.19; 79.2%), post-ischemic stroke weakness (n.4; 16.6%) and amyotrophic lateral sclerosis (n.1; 4.2%).

No significant differences in terms of sex (p-value 0.256), age (p-value 0.308), education (p-value 0.765) and disease duration (p-value 0.553) were found comparing patients with FMDs and patients with OND.

No significant differences in cognitive performances were recorded when comparing the two groups (**Table 1**).

At the Rorschach, FMDs presented a significantly higher frequency of Popular (P) and Sum of all Human content codes (SumH>5) responses than OND (For more details pertaining the variables, see S1 Table and Fig 1).

Moreover, the avoidant type of coping was significantly more frequent among FMDs than OND. At multivariate analysis, the associations were confirmed for both P responses (OR 5.78;

**Table 1. Neuropsychological assessment.**

| | FMD (n.31) | OND (n.24) | OR | 95% CI | p-value |
|---|---|---|---|---|---|
| MMSE score | 27.5±2.3 | 28.3±1.1 | 0.7 | 0.48–1.14 | 0.180 |
| MoCA score | 24.9±3.1 | 25.7±2.0 | 0.9 | 0.69–1.09 | 0.252 |
| FAB score | 15.6±1.0 | 15.8±1.0 | 0.9 | 0.58–1.32 | 0.541 |
| RAVLT immediate recall | 38.7±7.3 | 39.4±6.7 | 0.9 | 0.91–1.06 | 0.687 |
| RAVLT delayed recall | 7.8±2.1 | 7.4±1.8 | 1.1 | 0.84–1.49 | 0.406 |
| Stroop test, time (sec) | 26.7±8.6 | 26.3±8.8 | 1.0 | 0.94–1.07 | 0.839 |
| Stroop test, errors | 0.6±1.2 | 0.4±0.7 | 1.2 | 0.68–2.00 | 0.569 |
| Verbal fluency letter test (FAS) score | 30.8±9.7 | 33.3±11.1 | 0.9 | 0.92–1.02 | 0.375 |
| Clock drawing test | 0.9±1.4 | 0.6±1.2 | 1.1 | 0.75–1.77 | 0.514 |
| HAM-D score | 7.7±4.6 | 7.4±4.0 | 1.0 | 0.89–1.15 | 0.800 |
| STAY-1 score | 41.1±12.6 | 42.0±10.7 | 0.9 | 0.94–1.03 | 0.773 |
| STAY-2 score | 39.1±10.7 | 40.6±10.4 | 0.9 | 0.93–1.03 | 0.598 |

Data are expressed as mean and standard deviation. *Abbreviation*: MMSE: Mini Mental State Examination; MoCA: Montreal Cognitive Assessment; FAB: Frontal Assessment Battery; RAVLT: Rey's Auditory Verbal Learning Test; Verbal fluency letter test (FAS); CDT: Clock Drawing Test; HAM-D: Hamilton Depression Rating Scale; STAI Y1-Y2: State Trait Anxiety Inventory.

95% CI 1.04–31.94; p-value 0.044), SumH>5 (OR 11.74; 95% CI 1.88–73.20; p-value 0.008) and avoidant type of coping (OR 4.6; 95% CI 1.38–15.32; p-value: 0.013) (**Table 2**).

Concerning FMD phenotypes, SumY>2 was significantly more frequent in FMDs-wT (n.4, 50%) than FMDs-wGD (n.3, 13%; p-value: 0.031). Moreover, the introvertive and avoidant types of coping, as well as Zd<-3 and An+Xy>1 were more frequent in FMD-wT than FMDs-wGD, although the difference was not statistically significant (**Table 3**).

At the SCID-II, no statistically significant differences were recorded comparing the two groups. In particular, 8 (25.8%) patients with FMD and 3 (12.5%) patients with OND fulfilled the DSM-5 diagnostic criteria for PeDs (p-value 0.336). Out of the 8 FMD patients with PeD, 2 (25%) had obsessive-compulsive PeD, 2 (25%) had borderline PeD, 2 (25%) had avoidant PeD, 1 (12.5%) had dependent PeD and 1 (12.5%) had paranoid PeD. Out of the 3 patients with OND, 1 (33.3%) had obsessive-compulsive PeD, 1 (33.3%) had narcissistic PeD and 1 (33.3%) had paranoid PeD.

## Discussion

The Rorschach test has been previously used to assess personality functioning in patients suffering from neurological disorders, including multiple sclerosis, migraine and epilepsy [13,14]. However, to the best of our knowledge, to date no studies evaluating the personality structure of patients with FMDs applying a Rorschach approach are available.

In the present study, patients with FMDs presented peculiar personality characteristics compared to patients with OND. More in details, patients with FMDs presented a significantly higher percentage of P responses than patients with OND. Remarkably, P responses strictly reflect the individual ability to get involved in popular thinking, "social standards" and conformity. The "conformity" behavior, as part of the social implicit expectations, deeply affects one's own social attitudes leading to a voluntary behavioral change to imitate peers' behavior and respond to normative rules [15]. Interestingly, previous studies have hypothesized that the strict adherence to social compliance and conformity may represent a compensatory strategy

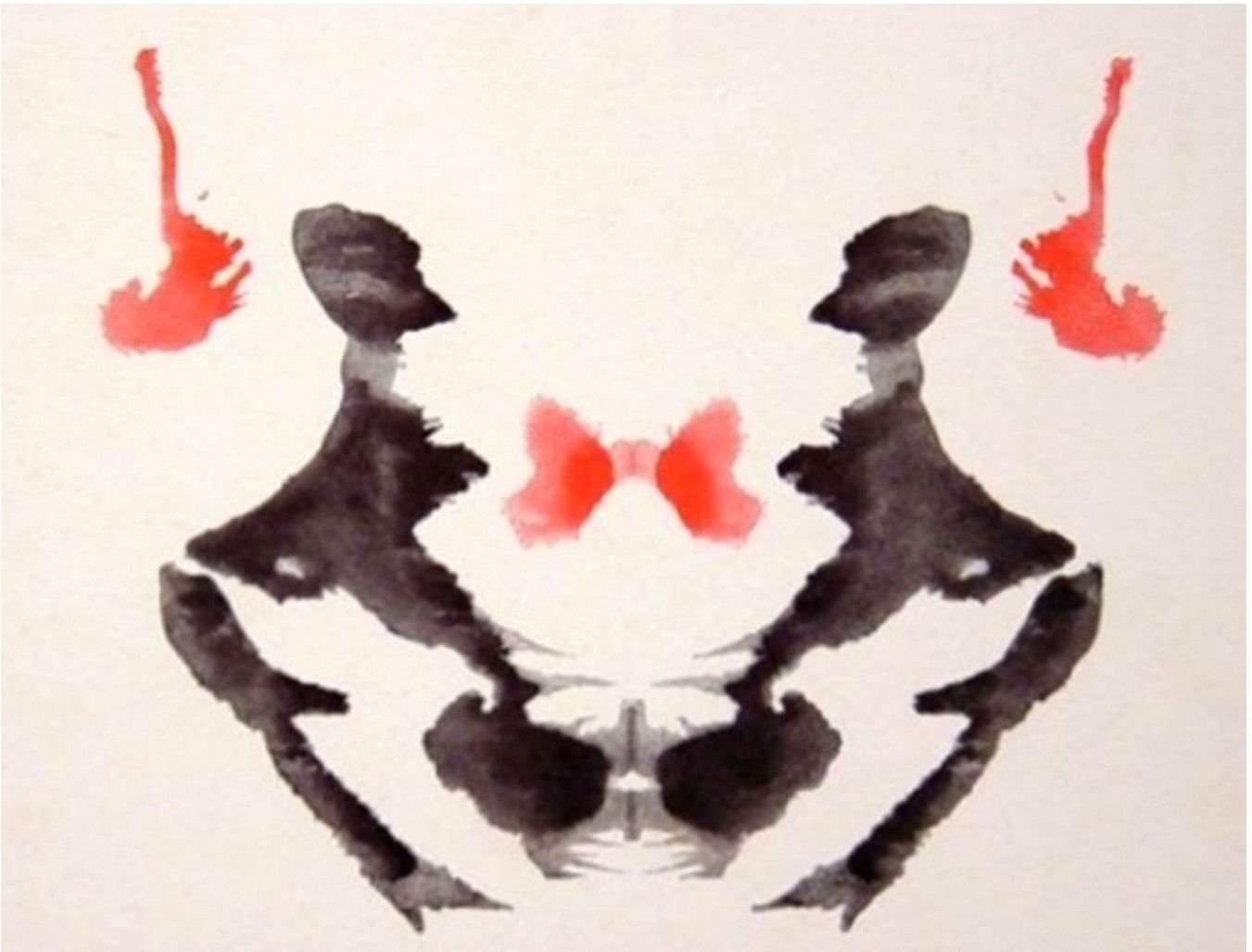

**Fig 1. The third card of the Rorschach test.** The most frequent (Popular) response is "Two humans interacting" [12].

to cope the private information insufficiency in case of high-ambiguity situation, thus assuming the "best behavior" [16].

Similarly, also the SumH variable was significantly higher in patients with FMDs than in patients with OND. Considering that the SumH variable represents the interest of the individual in others, the high percentage of SumH could reflect an "exaggerated" interest in others as the result of a higher than is typical innate drive for social acceptance, fearing of being excluded from the group and social pressures [17].

Pertaining phenotypes, FMDs-wT presented a significantly higher number of SumY responses than FMDs-wGD, thus reflecting "emotional confusion" and negative feelings stress-related. Moreover, respect to FMDs-wGD, patients with FMDs-wT presented a higher number of anatomy (i.e. liver, bones. . .) and X-Ray (i.e. X-ray of a pelvis, ultrasound of a fetus . . .) responses (An+Xy), classically more frequent among people with an everyday confrontation with body due to job position (i.e. doctors, nurses. . .) [18] but also in people suffering from somatizations and body concerns.

When PeDs diagnosed according DSM-5 criteria was assessed, no differences were found between FMDs and OND patients. Our findings are in line with the study performed by Defazio

**Table 2. Personality traits: The Rorschach approach.**

| | FMDs (n.31) | OND (n.24) | OR | 95% CI | p-value |
|---|---|---|---|---|---|
| **TYPE OF COPING** | | | | | |
| Introversive | 11 (35.5) | 9 (37.5) | 0.9 | 0.30–2.77 | 0.878 |
| Extratensive | 7 (22.6) | 3 (12.5) | 2.0 | 0.46–8.91 | 0.343 |
| Ambient | 13 (41.9) | 12 (50) | 0.7 | 0.24–2.19 | 0.552 |
| Avoidant | **20 (64.5)** | **7 (29.2)** | **4.4** | **1.40–13.90** | **0.011** |
| **CONTROL and STRESS TOLERANCE** | | | | | |
| AdjD<0 | 12 (38.7) | 8 (33.3) | 1.3 | 0.41–3.85 | 0.681 |
| CDI>3 | 13 (41.9) | 9 (37.5) | 1.2 | 0.40–3.58 | 0.739 |
| SumY>2 | 7 (22.6) | 7 (29.2) | 0.7 | 0.20–2.39 | 0.579 |
| FC<CF+C | 13 (41.9) | 13 (54.2) | 0.6 | 0.20–1.78 | 0.369 |
| **IDEATION** | | | | | |
| FM<2 | 13 (41.9) | 9 (37.5) | 1.2 | 0.40–3.58 | 0.739 |
| FM>5 | 1 (3.2) | 2 (8.3) | 0.4 | 0.03–4.30 | 0.425 |
| SumC'>2 | 7 (22.6) | 2 (8.3) | 3.2 | 0.60–17.12 | 0.172 |
| SumC'>WSumC | 12 (38.7) | 7 (29.2) | 1.5 | 0.49–4.79 | 0.462 |
| SumT>1 | 3 (9.7) | 3 (12.5) | 0.7 | 0.13–4.09 | 0.740 |
| SumV>0 | 13 (41.9) | 6 (25.0) | 2.2 | 0.67–6.96 | 0.194 |
| **INFORMATION PROCESSING** | | | | | |
| Zd<-3 | 26 (83.9) | 17 (70.8) | 2.1 | 0.58–7.85 | 0.251 |
| **COGNITIVE MEDIATION** | | | | | |
| XA%<0.70 | 7 (22.6) | 8 (33.3) | 0.6 | 0.17–1.92 | 0.377 |
| X-%>0.25 | 11 (35.5) | 8 (33.3) | 1.1 | 0.35–3.38 | 0.860 |
| P>6 | **10 (32.3)** | **2 (8.3)** | **5.2** | **1.02–26.77** | **0.033** |
| **AFFECT** | | | | | |
| DEPI>5 | 5 (16.13) | 4 (16.7) | 0.9 | 0.22–4.05 | 0.957 |
| HVI | 1 (3.2) | 2 (8.3) | 0.4 | 0.03–4.30 | 0.425 |
| **SELF-PERCEPTION** | | | | | |
| Fr+Rf>0 | 11 (35.5) | 5 (20.8) | 2.1 | 0.61–7.14 | 0.240 |
| **INTERPERSONAL PERCEPTION** | | | | | |
| SumH<3 | 7 (22.6) | 5 (20.8) | 1.1 | 0.61–7.14 | 0.876 |
| SumH>5 | **13 (41.9)** | **2 (8.3)** | **7.9** | **1.58–39.89** | **0.012** |
| An+Xy>1 | 17 (54.8) | 14 (58.3) | 0.9 | 0.29–2.54 | 0.796 |
| GHR<PHR | 4 (12.9) | 7 (29.2) | 0.3 | 0.09–1.41 | 0.144 |

Data are expressed as frequency and percentage. *Abbreviations*: FMD: Functional movement disorders; OND: Other neurological disorders; OR: Odds ratio; CI: Confidence intervals; Adj D: Adjusted Difference Score. CDI: Coping Deficit Index; SumY: Sum of diffuse shading determinants; FC: Form-color; CF: Color-form; C: Color; FM: Animal Movement; SumC': Sum of color; WSumC (Weighted Sum of Color); SumT: Sum of texture determinants; SumV: Sum of vista determinants; Zd: Efficiency index; XA%: Accuracy in perception; X-%; distortion in perception; P: Popular; DEPI: Depression index; HVI: Hypervigilance Index; Fr: Rf: SumH: Sum human; An: Anatomy; Xy: X-ray; GHR: Good human representational responses; PHR: Poor human representational responses. Bold data: p-value<0.005.

and coll. [19], which did not find any differences in the frequency of PeDs between patients with FMDs and patients with OND. Nevertheless, as for the study performed by Defazio and coll. [19], due to the small sample size, it could not be excluded that the non-significant difference recorded between FMDs and OND in PeDs frequency could be the result of a low statistical power.

Moreover, in our sample, FMDs frequently presented an avoidant style of coping, characterized by a maladaptive way in which they deal with problems. Consequently, patients with FMDs may isolated themselves from interpersonally complex situations, applying defensive

**Table 3. Personality traits according to FMD phenotypes.**

| | FMDs-wT (n.8) | FMDs-GD (n.23) | p-value |
|---|---|---|---|
| **TYPE OF COPING** | | | |
| *EB* | | | |
| EB-Introversive | 3 (37.5) | 7 (30.4) | 0.562 |
| EB-Extratensive | 2 (25) | 5 (21.7) | 0.599 |
| EB-Ambitent | 3 (37.5) | 11 (47.8) | 0.483 |
| *Avoidant* | 6 (75) | 14 (60.9) | 0.788 |
| *CONTROL and STRESS TOLLERANCE* | | | |
| AdjD<0 | 3 (37.5) | 9 (39.1) | 0.935 |
| CDI>3 | 2 (25) | 11 (47.8) | 0.260 |
| SumY>2 | 4 (50) | 3 (13.0) | **0.031** |
| FC<CF+C | 3 (37.5) | 10 (43.5) | 0.768 |
| **IDEATION** | | | |
| FM<2 | 3 (37.5) | 10 (43.5) | 0.768 |
| FM>5 | 1 (12.5) | 1 (4.3) | 0.258 |
| SumC'>2 | 0 | 7 (30.4) | / |
| SumC'>WSumC | 3 (37.5) | 9 (39.1) | 0.935 |
| SumT>1 | 1 (12.5) | 2 (8.7) | 0.754 |
| SumV>0 | 3 (37.5) | 10 (43.5) | 0.768 |
| **INFORMATION PROCESSING** | | | |
| Zd<-3 | 8 (100) | 18 (78.3) | 0.150 |
| **COGNITIVE MEDIATION** | | | |
| XA%<0.70 | 2 (25) | 5 (21.7) | 0.841 |
| X-%>0.25 | 3 (37.5) | 8 (34.8) | 0.890 |
| P>6 | 3 (37.5) | 7 (30.4) | 0.725 |
| **AFFECT** | | | |
| DEPI>5 | 0 | 5 (21.7) | / |
| HVI | 1 (12.5) | 0 | / |
| **SELF-PERCEPTION** | | | |
| Fr+Rf>0 | 3 (37.5) | 8 (34.7) | 0.890 |
| **INTERPERSONAL PERCEPTION** | | | |
| SumH<3 | 1 (12.5) | 6 (26.1) | 0.493 |
| SumH>5 | 3 (37.5) | 10 (43.5) | 0.793 |
| An+Xy>1 | 5 (62.5) | 12 (52.2) | 0.646 |
| GHR<PHR | 0 | 4 (17.4) | / |

Data are expressed as frequency and percentage. *Abbreviations*: FMD: Functional movement disorders; wT: Weakness and tremor; wGD: Weakness and gait disorders; CDI: Coping Deficit Index; SumY: Sum of diffuse shading determinants; FC: Form-color; CF: Color-form; C: Color; FM: Animal Movement; SumC': Sum of color; WSumC (Weighted Sum of Color); SumT: Sum of texture determinants; SumV: Sum of vista determinants; Zd: Efficiency index; XA%: Accuracy in perception; X-%; distortion in perception; P: Popular; DEPI: Depression index; HVI: Hypervigilance Index; Fr: Rf: SumH: Sum human; An: Anatomy; Xy: X-ray; GHR: Good human representational responses; PHR: Poor human representational responses. Bold data: p-value<0.005.

maneuvers to "protect" themselves by the challenges of everyday life. Indeed, recent studies have reported that FMDs presented more frequently an insecure and avoidant attachment style [20], characterized by the reluctance to self-disclose and emotional suppression used as a strategy to cope with fear of judgment and rejection.

Summarizing, in the present study patients with FMDs presented a personality dimension characterized by "conformity behaviors" and excessive interest in others than usual and a

maladaptive avoidant style of coping. These psychopathological characteristics may favor the occurrence of FMDs that mimic organic movement disorders, probably less stigmatized, misunderstood and victim of harmful social distancing than non-organic ones.

The scoring of the Rorschach inkblot test is undoubtedly complex, time-consuming and painstaking. However, our findings support the usefulness of the Rorschach approach in identifying personality "weaknesses and strengths" that should be considered in the treatment planning. Furthermore, in our study, personality traits were assessed using the multilevel personality assessment framework (SCID-II, Rorschach test, comprehensive neuropsychological battery) that, notably, might explain possible personality-based underpinnings of the development and maintenance of FMD with possible research, clinical and treatment implications.

The present study has some limitation. The enrolment of FMDs mainly characterized by functional weakness may reduce the generalizability of the findings. Moreover, although about 80% of OND were represented by patients with multiple sclerosis, a bias related to the heterogeneity of the OND group could not be ruled out. Furthermore, the small sample size and the large 95% CI recorded did not allow us to exclude that the lack of significant differences recorded in several Rorschach variables could be the result of a low statistical power. However, more patients than those required by the sample size calculation were enrolled. Finally, although the Rorschach test provides valuable insights into personality characteristics, it could be subject to interpretation across different raters. Nevertheless, to reduce possible bias, all the enrolled patients were assessed by the same rater, whose experience in Rorschach administration and coding is certified.

In conclusion, our findings may be useful not only to better understand predisposing factors of FMDs but also to build a "good" therapeutic alliance, necessary to plan a personalized treatment, frequently requiring an integrated multidisciplinary approach [21]. Indeed, the assessment of personality characteristics may be useful to maximize the adherence, reducing "beliefs" that may represent barriers, enhancing the readiness for change and, ultimately, improving treatment success.

Larger studies including healthy control group matched by age, sex and education are needed to confirm our finding and to define the possible role of personality not only in FMDs occurrence but also in determining its presentation with one clinical phenotype (i.e. tremor, dystonia, gait disorders, parkinsonism, chorea. . .) rather than another.

## Supporting information

**S1 Checklist. The RECORD statement–checklist of items, extended from the STROBE statement, that should be reported in observational studies using routinely collected health data.** (DOCX)

**S1 Table. Description and interpretation of the Rorschach variables.** Abbreviations: Adj D: Adjusted Difference Score. CDI: Coping Deficit Index; SumY: Sum of diffuse shading determinants; FC: Form-color; CF: Color-form; C: Color; FM: Animal Movement; SumC': Sum of color; WSumC (Weighted Sum of Color); SumT: Sum of texture determinants; SumV: Sum of vista determinants; Zd: Efficiency index; XA%: Accuracy in perception; X-%; distortion in perception; P: Popular; DEPI: Depression index; HVI: Hypervigilance Index; Fr: Rf: SumH: Sum human; An: Anatomy; Xy: X-ray; GHR: Good human representational responses; PHR: Poor human representational responses. (DOCX)

**S1 Dataset.** (XLSX)

## Author Contributions

**Conceptualization:** Antonina Luca, Tiziana Lo Castro, Alessandra Nicoletti, Mario Zappia.

**Data curation:** Antonina Luca, Tiziana Lo Castro.

**Formal analysis:** Antonina Luca.

**Investigation:** Antonina Luca, Tiziana Lo Castro.

**Methodology:** Antonina Luca, Giovanni Mostile, Giulia Donzuso, Calogero Edoardo Cicero, Alessandra Nicoletti, Mario Zappia.

**Project administration:** Antonina Luca.

**Supervision:** Alessandra Nicoletti, Mario Zappia.

**Writing – original draft:** Antonina Luca.

**Writing – review & editing:** Antonina Luca, Tiziana Lo Castro, Giovanni Mostile, Giulia Donzuso, Calogero Edoardo Cicero, Alessandra Nicoletti, Mario Zappia.

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
