## [Decision Letter · Decision Letter 0]

27 Mar 2024

PONE-D-24-04506Personality and psychopathological characteristics in functional movement disordersPLOS ONE

Dear Dr. Zappia,

Thank you for submitting your manuscript to PLOS ONE. After careful consideration, we feel that it has merit but does not fully meet PLOS ONE’s publication criteria as it currently stands. Therefore, we invite you to submit a revised version of the manuscript that addresses the points raised during the review process.

Please follow carefully the instructions of both Reviewers and make all amendments according to their suggestions. Moreover, all the text should be revised for typos (e.g., line 101 for undue comma, line 107 for missing dot, line 125 for missing space...). In statistical analyses, it should be specified whether the shape of distribution of variables was studied, and therefore why a parametric test was chosen (t-Student).

We look forward to receiving your revised manuscript.

Kind regards,

Simone Varrasi

Academic Editor

PLOS ONE

Reviewers' comments:

Reviewer's Responses to Questions

**Comments to the Author**

1. Is the manuscript technically sound, and do the data support the conclusions?

Reviewer #1: Yes

Reviewer #2: Yes

2. Has the statistical analysis been performed appropriately and rigorously? 

Reviewer #1: Yes

Reviewer #2: Yes

3. Have the authors made all data underlying the findings in their manuscript fully available?

Reviewer #1: Yes

Reviewer #2: Yes

4. Is the manuscript presented in an intelligible fashion and written in standard English?

Reviewer #1: Yes

Reviewer #2: Yes

5. Review Comments to the Author

Reviewer #1: The work presented by the authors aimed to compare the personality and psychopathological characteristics of patients with functional movement disorders (FMDs) and other neurological disorders (OND). The authors highlight that FMDs patients showed a higher frequency of certain responses on the Rorschach test and exhibited avoidant coping compared to OND patients. These findings suggest potential psychopathological traits associated with FMDs.

I think the work is structured and well-written. I only have to propose some suggestions to increase the quality of the work.

Including a figure illustrating an example of a "popular response" from the Rorschach test would be beneficial. Such visual representation could offer clarity and deepen understanding for readers.

I suggest specifying whether the Rorschach test has been used in other neurological populations. This additional information could provide valuable context and broaden the scope of the findings.

In the discussion section, I recommend emphasizing the importance of the Rorschach test despite its length and specialized nature. Stressing how it provides relevant information for therapeutic decision-making could underscore its clinical utility.

Reviewer #2: This is a well written study on the personality characteristics of patients with Functional Movement Disorders. Using the Rorschach test and comparing results with a cohort pf patients affected by other neurological disorders (mostly MS), the authors showed a higher prevalence of conformity behaviours and of an avoidant style among individuals with FMD. While the small sample size stands as its primary limit, this study offers a further valuable insight to disentangle the pathological basis of such disorders.

I have no particular comments.

As a small suggestion, I was wondering whether it could be useful, for future studies, to also include a healthy control group, matched for relevant demographic variables.

6. PLOS authors have the option to publish the peer review history of their article (what does this mean?). If published, this will include your full peer review and any attached files.

Reviewer #1: No

Reviewer #2: No

---

## [Author Response · Author response to Decision Letter 0]

18 Apr 2024

Dear Editor,

We would like to thank you and the reviewers for appreciating our study. 

Please, find the point-by-point responses to the Reviewers’ comments that, in our opinion, improved the quality of the manuscript.

Reviewer #1: The work presented by the authors aimed to compare the personality and psychopathological characteristics of patients with functional movement disorders (FMDs) and other neurological disorders (OND). The authors highlight that FMDs patients showed a higher frequency of certain responses on the Rorschach test and exhibited avoidant coping compared to OND patients. These findings suggest potential psychopathological traits associated with FMDs.

I think the work is structured and well-written. I only have to propose some suggestions to increase the quality of the work.

Including a figure illustrating an example of a "popular response" from the Rorschach test would be beneficial. Such visual representation could offer clarity and deepen understanding for readers.

We thank the reviewer for her/his suggestion. Accordingly, we added the third card of the Rorschach test as example of Popular response (Fig.1).

I suggest specifying whether the Rorschach test has been used in other neurological populations. This additional information could provide valuable context and broaden the scope of the findings.

We thank the reviewer for the suggestion. We have added a sentence in the discussion, specifying that the Rorschach test has been used also in patients with multiple sclerosis, migraine and epilepsy (lines 232-234).

In the discussion section, I recommend emphasizing the importance of the Rorschach test despite its length and specialized nature. Stressing how it provides relevant information for therapeutic decision-making could underscore its clinical utility.

We thank the reviewer. We added a paragraph in the discussion section underlying the usefulness of the Rorschach test in the treatment planning despite its complex, time-consuming and painstaking nature (lines 273-275).

Reviewer #2: This is a well written study on the personality characteristics of patients with Functional Movement Disorders. Using the Rorschach test and comparing results with a cohort of patients affected by other neurological disorders (mostly MS), the authors showed a higher prevalence of conformity behaviours and of an avoidant style among individuals with FMD. While the small sample size stands as its primary limit, this study offers a further valuable insight to disentangle the pathological basis of such disorders.

I have no particular comments. As a small suggestion, I was wondering whether it could be useful, for future studies, to also include a healthy control group, matched for relevant demographic variables.

We are grateful to the reviewer for the positive evaluation of our paper. As suggested, we added a sentence in the discussion section underlying the usefulness of future studies including healthy controls to confirm our findings (294-295).

Moreover, as suggested by the Editor and the editorial office, we revised the manuscript for typos and we specified, in the methods section, the statistical analysis performed to assess the variables distribution (lines 122-125). Furthermore, we ensured that the manuscript meets PLOS ONE's style requirements, we delated the ethics statement from the title page and we included captions for the Supporting Information file at the end of the manuscript.

---

## [Editor Report · Decision Letter 1]

24 Apr 2024

Personality and psychopathological characteristics in functional movement disorders

PONE-D-24-04506R1

Dear Dr. Zappia,

We’re pleased to inform you that your manuscript has been judged scientifically suitable for publication and will be formally accepted for publication once it meets all outstanding technical requirements.

Kind regards,

Simone Varrasi

Academic Editor

PLOS ONE
---

## [Editor Report · Acceptance letter]

29 Apr 2024

PONE-D-24-04506R1 

PLOS ONE

Dear Dr. Zappia, 

I'm pleased to inform you that your manuscript has been deemed suitable for publication in PLOS ONE. Congratulations! Your manuscript is now being handed over to our production team.

Kind regards, 

on behalf of

Dr. Simone Varrasi 

Academic Editor

PLOS ONE